# Imaging correlates of visual function in multiple sclerosis

**Eduardo Caverzasi**[1☯], **Christian Cordano**[1☯], **Alyssa H. Zhu**[2], **Chao Zhao**[1],
**Antje Bischof**[1,3], **Gina Kirkish**[1], **Daniel J. Bennett**[1], **Michael Devereux**[1], **Nicholas Baker**[1],
**Justin Inman**[1], **Hao H. Yiu**[1], **Nico Papinutto**[1], **Jeffrey M. Gelfand**[1], **Bruce A. C. Cree**[1],
**Stephen L. Hauser**[1], **Roland G. Henry**[1], **Ari J. Green**[1,4]*

1 Division of Neuroimmunology and Glial Biology UCSF, Weill Institute for Neurosciences, Department of Neurology, University of California, San Francisco, San Francisco, CA, United States of America, 2 Imaging Genetics Center, Stevens Neuroimaging and Informatics Institute, University of Southern California, United States of America, 3 Neurology and Immunology Clinic, University Hospital Basel, Switzerland, 4 Department of Ophthalmology, University of California, San Francisco, San Francisco, CA, United States of America

☯ These authors contributed equally to this work.
* agreen@ucsf.edu

**Data Availability Statement:** All relevant data are within the manuscript and its Supporting Information files.

**Funding:** Author CC was supported by a Training Fellowship FISM (Fondazione Italiana Sclerosi

## Abstract

No single neuroimaging technique or sequence is capable of reflecting the functional deficits manifest in MS. Given the interest in imaging biomarkers for short- to medium-term studies, we aimed to assess which imaging metrics might best represent functional impairment for monitoring in clinical trials. Given the complexity of functional impairment in MS, however, it is useful to isolate a particular functionally relevant pathway to understand the relationship between imaging and neurological function. We therefore analyzed existing data, combining multiparametric MRI and OCT to describe MS associated visual impairment. We assessed baseline data from fifty MS patients enrolled in ReBUILD, a prospective trial assessing the effect of a remyelinating drug (clemastine). Subjects underwent 3T MRI imaging, including Neurite Orientation Dispersion and Density Imaging (NODDI), myelin content quantification, and retinal imaging, using OCT. Visual function was assessed, using low-contrast letter acuity. MRI and OCT data were studied to model visual function in MS, using a partial, least-squares, regression analysis. Measures of neurodegeneration along the entire visual pathway, described most of the observed variance in visual disability, measured by low contrast letter acuity. In those patients with an identified history of ON, however, putative myelin measures also showed correlation with visual performance. In the absence of clinically identifiable inflammatory episodes, residual disability correlates with neurodegeneration, whereas after an identifiable exacerbation, putative measures of myelin content are additionally informative.

## Introduction

Functional impairment can take a long time to manifest in multiple sclerosis (MS) [1, 2]. Shorter-term clinical trials are needed to assess therapeutic efficacy over reasonable time

Multipla) - Cod. 2013/B/4. AB received travel fees from Actelion and speaker fees from Biogen, and has received research support from the Freiwillige Akademische Gesellschaft, Switzerland, and the Gottfried and Julia Bangerter-Rhyner Stiftung, Switzerland. JMG received research support via UCSF from Genentech for a clinical trial and consulting for Biogen and Alexion. In the past 36 months, BACC has received personal compensation for consulting from Akili, Alexion, Atara, Biogen, EMD Serono, Novartis, Sanofi and TG Therapeutics. SLH was supported by funding from the National Institute of Neurological Disorders and Stroke (R35NS111644) and the Valhalla Foundation. AJG was supported by funding from University of California, San Francisco and the Rachleff Family. The funders had no role in study design, data collection and analysis, decision to publish, or preparation of the manuscript. The specific roles of these authors are articulated in the 'author contributions' section.

**Competing interests:** The authors have read the journal's policy and have the following potential competing interests: SLH serves on the Board of Directors for Neurona, on the Scientific Advisory Board for Alector, Annexon, Bionure, and Molecular Stethoscope, and he has also received non-financial support from F. Hoffmann-La Roche Ltd and Novartis AG. AB has received travel fees from Actelion and speaker fees from Biogen. JMG reports research support to UCSF from Genentech for a clinical trial and consulting for Biogen and Alexion. BACC has received personal compensation for consulting from Akili, Alexion, Atara, Biogen, EMD Serono, Novartis, Sanofi and TG Therapeutics. AJG has served on the Scientific Advisory Board for Bionure, Inception Sciences and Pipeline Therapeutics. He serves as an Associate Editor at JAMA Neurology. He has served on an Adjudication Committee for MedImmune/Viela Bio. He has provided expert support for Mylan, Synthon and Pharmasciences and personal fees from and other financial relationships with Pipeline Therapeutics. This does not alter our adherence to PLOS ONE policies on sharing data and materials. There are no patents, products in development or marketed products to declare. All other authors have declared that no competing interests exist.

scales. Imaging methods have been employed for this purpose [3, 4], but the correlation between imaging metrics and functional impairment remains uncertain and the specific imaging techniques that are best suited for measurement have not yet been defined [5, 6]. This is particularly important for investigations of neuroprotective and reparative therapies.

The visual pathway is anatomically well defined and functionally discrete, with well-validated measures that are both quantifiable and reliable. For these reasons, the visual pathway has long been used as a model for understanding pathological processes (e.g. inflammation, demyelination, and neurodegeneration) that underlie neurological dysfunction in the disease [7, 8].

ReBUILD, a double-blind, placebo-controlled trial, demonstrated the effectiveness of clemastine fumarate in improving visual-evoked, potential latency and visual function [9]. To further explore the relationship between imaging and function, and in particular to overcome the mismatch between conventional MRI and clinical deficit in MS [6], we performed an analysis based on optical coherence tomography (OCT) and advanced MRI assessments from the baseline assessment of the ReBUILD trial cohort [9]. To describe visual impairment in MS, we specifically implemented a multimodal approach, combining OCT and novel MRI techniques capable of estimating both metrics for myelin content, such as myelin water fraction (MWF) [10, 11] and metrics for neurite organization (neurite density (NDI) and orientation dispersion indexes (ODI)) [12].

## Methods

### Subjects and clinical evaluation

Fifty relapse-remitting (RRMS) patients underwent baseline assessments as part of the ReBUILD trial prior to the initiation of treatment (32 females, average age of 40.1 years ± 10.3 SD, median EDSS 2.0 (range 0 to 5.5), and median disease duration of 3.2 years (range 0.4 to 30.4). Either a clinically evident, optic neuritis that had occurred within 6 months before screening or more than 5 years prior to enrollment in the qualifying eye for the Rebuild study was an exclusion criterion. For the purpose of this study we considered all patients' eyes for which data was available, including those eyes with ON older than 5 years (i.e. eyes that were not basis for qualification in the study). Visual impairment was measured by low contrast letter acuity (LCLA; Sloan 2.5% low contrast vision chart; Precision Vision, La Salle, IL, USA). The single eye measures were averaged with the fellow eye ones to have a single value for each patient. The study was approved by the UCSF Institutional Review Board and all participants provided informed consent. The trial was registered at ClinicalTrials.gov (number NCT02040298) before initiation of patient enrolment. All research was performed in accordance with relevant guidelines/regulations.

### MRI acquisition

Each subject underwent brain MRI acquisitions on a 3T Siemens Skyra scanner. The MRI protocol included standard sagittal T1-weigthed, 3D MPRAGE (1 mm$^3$ cubic voxel), a two-shell, neurite-orientation dispersion and density imaging (NODDI) [12] protocol (30 & 64 directions at b = 700 & 2000 s/mm$^2$, 2.2 mm$^3$ cubic voxel) and a multi-echo gradient-echo (MEGE) sequence for myelin content quantification [10, 11]. Details of the MRI protocol were previously reported [9].

## MRI analysis

Automated parcellation of T1-weighted volumes was performed using Freesurfer Image Analysis Suite Version 5.6. Total cortex, primary visual area (V1, Brodman area 17), thalami and cerebellar cortex were parcellated as volumes of interest (VOIs) from T1-weigthed, 3D MPRAGE image (Fig 1) [13–15]. Computed volumes were normalized by intracranial volume. Optic radiation (OR) VOIs were obtained from the JHU DTI-based (MNI space) white-matter atlases [16] and co-registered to the single subject space using linear and nonlinear transformations (FLIRT/FNIRT) [17, 18]. After correcting for distortions due to eddy current and head motion, maps of mean diffusivity (MD) were calculated by fitting the diffusion tensor model within each voxel, using dipy fit tensor [19]. The NODDI model was fitted to the diffusion datasets in MATLAB (http://mig.cs.ucl.ac.uk/Tutorial.NODDImatlab) and maps of ODI were therefore computed [12]. MEGE data were instead processed to obtain quantification of the MWF [10, 11]. MD, ODI and MWF maps were co-registered to T1 space, using linear and nonlinear transformations (FLIRT/FNIRT) [17, 18]. The mean value of each MRI metric was calculated for each VOI (V1, thalami and cerebellar cortex) averaging left and right hemispheres.

**Lesion burden.** An expert neuroradiologist (EC) assessed the number of occipital cortical lesions and also segmented the total white-matter-lesion burden within the OR, using the available FLAIR and T1-weigthed 3D MPRAGE images. We confirmed the location of lesions by using occipital lobes masks derived by Freesurfer and JHU DTI-based OR registered in the single subject space.

## OCT

Retinal imaging was performed using spectral-domain OCT (Heidelberg Engineering, Heidelberg, Germany, eye explorer software version 1.9.10.0) as previously described [7].

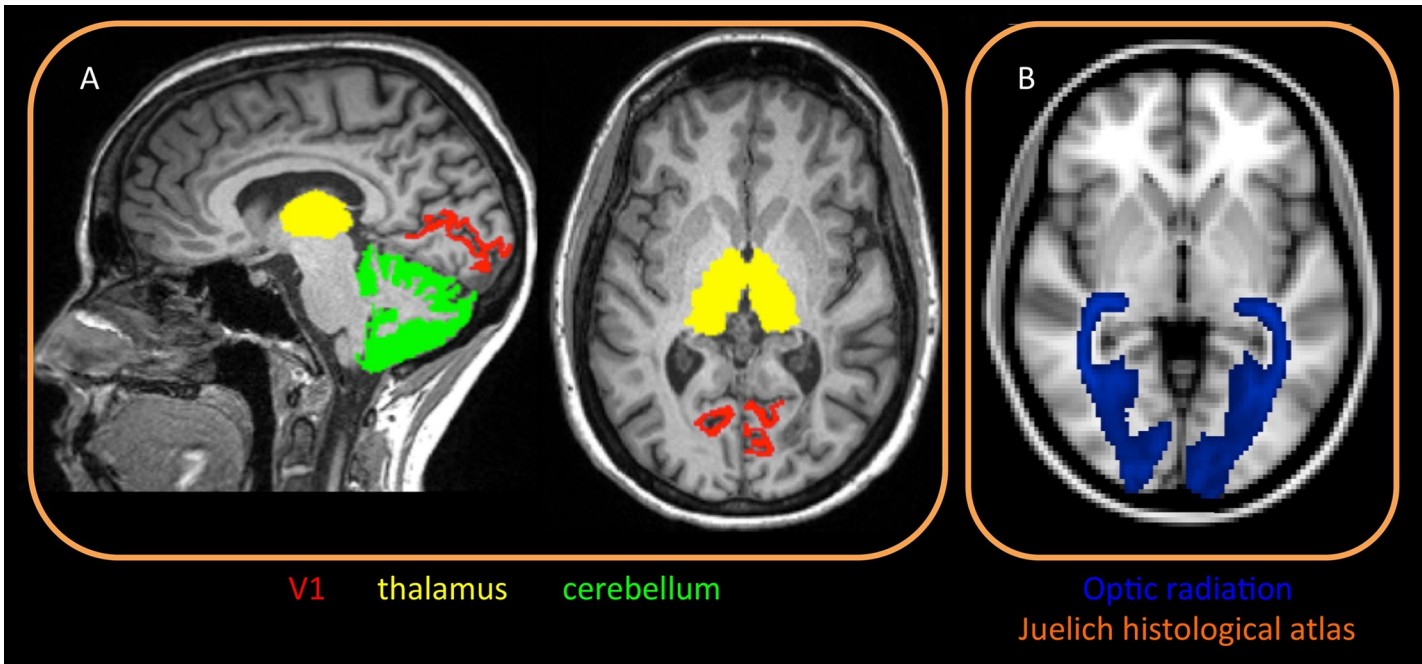

**Fig 1. MRI analysis methods: Volume of interests.** A: Freesurfer pipeline was used to segment specific gray matter regions belonging to the visual network, specifically thalamus, cerebellar cortex and V1 (primary visual area). Region of interest were identified on each subject. B: probabilistic map of optic radiation from the Juelich histological atlas (on MNI space).

Peripapillary, retinal-nerve-fiber-layer (pRNFL) thickness and macular volume (MV) were obtained. Intra-retinal-layer segmentation was executed to quantify the Ganglion Cell Layer (GCL) through the Viewing Module 6.0. We followed the APOSTEL guidelines for reporting OCT studies (see S1 and S2 Tables) [20].

**Statistical analysis.** Using a two-tailed t-test, we tested comparisons of demographic characteristics between ON negative and ON positive groups. To describe the variability in MS-associated visual impairment (LCLA), we performed a PLS to model baseline LCLA. using demographic (age at disease onset, and at MRI), MRI (normalized volumes, MD, myelin, ODI) and OCT metrics (GCL, pRNFL, MV), separately considering negative and positive history of ON. The significance of each predictor was established once the "*Variable Importance in the Projection*" (VIP) parameter was greater than 0.8 [21]. Random-forest approach was also used to test the reliability of the variable selected in the model. K-fold, cross-validation analysis was performed to test the reliability of each identified model by 4-fold cross-validation with 100 repeats between least absolute shrinkage and selection operator (LASSO) and PLS. The predictive accuracy was measured by root-mean-square error (RMSE) on both training data and test data. Model selection, using PLS has the advantage of avoiding co-linearity-related data inflation, although this approach lacks an efficient method for subset selection. LASSO was therefore applied to confirm the importance of the identified variables.

## Results

Twenty-eight patients had a prior history of optic neuritis (ON), whereas 22 did not. Demographic and clinical characteristics are reported in Table 1. LCLA scores for the 2 groups (22.9 ±10.1 and 23.1±9.2) were lower than the LCLA values for healthy controls in literature (45.48 ±11.22) [22]. There was a difference in age between the ON negative and ON positive groups (p <0.05). MRI and OCT metrics data are reported in S1 Table.

Using partial least squares regression analysis (PLS), we identified model predictors of LCLA with R-squared values up to 0.43 and 0.39 with or without ON respectively (Table 2). GCL proved to be the best OCT-related, partial predictor in particularly in cases of a negative history of ON (*Variable Importance in the Projection* (VIP) = 1.12 and 0.84 without and with previous ON, respectively) (S1 and S2 Tables). Lesion burden and OCT metrics appeared to be more informative for patients with no history of ON. Measures of cortical neurodegeneration seemed equally informative in both negative and positive history of optic neuritis, contributing to clinical disability. Thalamic volume was more informative in patients without ON.

**Table 1. Demographic and clinical characteristics of the groups with negative and positive history of optic neuritis (ON).**

| | ON negative history group | ON positive history group |
|---|---|---|
| **N#** | 22 | 28 |
| **Age** | 43.9 (9.7) | 37.1 (9.9) |
| **Sex** | 14 F (64%) | 18 F (64%) |
| **Disease duration (years)** | 3.5 (0.7 to 14.2) | 2.9 (0.45 to 30.4) |
| **Time from ON (years)** | / | 3.0 (0.2 to 15.8) |
| **EDSS** | 2.0 ± (0 to 5.5) | 2.0 (0 to 4) |
| **LCLA** | 22.9 (10.1) | 23.1 (9.2) |
| **VEP** | 125.7 (8.6) | 129.0 (11.4) |

Demographic data, LCLA abd VEP are reported as mean (SD) or n (%). Disease duration, time from ON and EDSS are reported at median (range).

Table 2. Schematic representations of the LCLA model results in subjects with negative and positive history of optic neuritis.

| LCLA MODEL | | Variable | VIP | Scale Coefficient |
|---|---|---|---|---|
| LCLA IN NEGATIVE HISTORY OF ON | R squared 42.8 | GCL | 1.12 | 0.2364 |
| | | OR lesion volume | 1.01 | -0.2123 |
| | | pRNFL | 1.00 | 0.2105 |
| | | Thalamic volumes | 0.93 | 0.1961 |
| | | V1 volume | 0.92 | 0.1929 |
| LCLA IN POSITIVE HISTORY OF ON | R squared 39.3 | Thalamic MWF | 1.19 | 0.22 |
| | | OR MWF | 1.11 | 0.21 |
| | | Cortical GM volume | 0.96 | 0.18 |
| | | V1 volume | 0.86 | -0.16 |
| | | GCL | 0.84 | 0.16 |

We reported the R squared of each model. The significance of each variable selected by PLS is reported as "Variable Importance in the Projection" (VIP). The scale coefficient, representative of the effect size for each variable, is also shown. ON = optic neuritis; MWF = myelin water fraction; GCL = Ganglion Cell Layer; OR = optic radiation; pRNFL = peripapillary retinal nerve fiber layer; GM = gray matter.

Measures of myelin content (MWF) within the OR and thalamus instead gained importance when we evaluated patients with ON. A random-forest approach confirmed all variables of the two models except for the whole-brain, cortical, gray matter in the ON positive history model.

## Model cross-validation analysis

PLS models showed low RMSE variability on both training data and test data, suggesting that they yielded stable predictions. Among the two PLS models, 9 out of 10 predictors selected by PLS also survived in the LASSO models' subset selection.

## Discussion

Evaluating the ReBUILD baseline data set, we identified MRI and OCT metrics-based models that showed a strong correlation with visual performance and demonstrated good reliability. Our results employ a more complete analysis of a single pathway because of the depth of data acquisition, the focus on myelin metrics and the capacity to assess structural injury to the visual system.

These data indicate that in patients with a history of ON, myelin loss is the main contributor to clinical, visual disability in MS. In this group, both MRI markers for putative myelin content (MWF) and neurodegeneration are significantly associated with visual function. By contrast, in patients without prior episodes of optic neuritis, axonal neurodegeneration appears to be responsible for most of the clinical disability, together with white matter lesions within the optic radiations.

There are different potential explanations for these observations. One is that separate processes underlie neurodegeneration and myelin damage; i.e. there is a degenerative process acting in MS that is independent of myelin loss. Alternatively, myelin loss and neuroaxonal loss could be time-disassociated (i.e. neurodegeneration may follow myelin loss by a prolonged period—even years - and therefore functional impairment from an asymptomatic episode in the more distant past may be more likely to be related to neuroaxonal loss). The differences between the models might also be due to damage in the posterior pathway (OR lesions, V1 neurodegeneration), contributing to visual disability particularly in patients without ON.

The strength of our models is their demonstrated robustness. A weakness of our study, however, is that some of the data may have been influenced by the criteria for inclusion in the

clinical trial [9]. For example, patients with mild visual disability or dysfunction were likely excluded.

We attempted to address limitations related to sample size and lack of an independent dataset for validation by using a nested cross-validation of the identified models. The size of the cohort also forced us to select a subset of the available MRI metrics before performing our analysis to minimize the risk of redundancy (e.g. using diffusion MRI ODI versus fractional anisotropy). This means that other similar metrics may perform similarly or better in future cohorts evaluating the same outcomes. Our failure to identify GM lesion burden as a meaningful factor might also be due to the study design that did not include dedicated cortical MRI sequences [23] and therefore resulted in poor sensitivity in detecting cortical lesions [24]. In order to be applied in observational and clinical studies, the obtained predictive models need to be replicated in larger populations.

In conclusion, we have demonstrated that combinations of advanced multiparametric MRI sequences within the visual network and OCT correlate with visual performance, adding to our understanding of the underlying pathological mechanisms responsible for clinical dysfunction. These measures should be candidates for observational and clinical studies in the visual pathways in MS, as our field shifts its focus on regenerative medicine.

## Supporting information

**S1 Table. Schematic representations of the LCLA model results looking at the entire cohort.** We reported the R squared. The significance of each variable selected by PLS is reported as "Variable Importance in the Projection" (VIP). The scale coefficient, representative of the effect size for each variable, is also shown. ON = optic neuritis; MWF = myelin water fraction; GCL = Ganglion Cell Layer; OR = optic radiation; pRNFL = peripapillary retinal nerve fiber layer; GM = gray matter.
(DOCX)

**S2 Table. OCT and MRI metrics values are reported for subject with negative and positive history of ON.** Data are mean (SD). Volumes are reported as ($mm^3$). MD $10^{-3}$ $mm^2$/s.
(DOCX)

**S1 Dataset.**
(XLSX)

## Author Contributions

**Conceptualization:** Eduardo Caverzasi, Christian Cordano, Nico Papinutto, Roland G. Henry, Ari J. Green.

**Data curation:** Eduardo Caverzasi, Christian Cordano, Alyssa H. Zhu, Daniel J. Bennett, Jeffrey M. Gelfand, Ari J. Green.

**Formal analysis:** Eduardo Caverzasi, Christian Cordano, Alyssa H. Zhu, Chao Zhao, Antje Bischof, Gina Kirkish, Daniel J. Bennett, Michael Devereux, Nicholas Baker, Justin Inman, Hao H. Yiu, Nico Papinutto.

**Methodology:** Christian Cordano, Gina Kirkish.

**Resources:** Chao Zhao.

**Supervision:** Bruce A. C. Cree, Stephen L. Hauser, Roland G. Henry, Ari J. Green.

**Validation:** Eduardo Caverzasi.

**Visualization:** Eduardo Caverzasi.

**Writing – original draft:** Eduardo Caverzasi, Christian Cordano, Alyssa H. Zhu, Chao Zhao, Antje Bischof, Nico Papinutto, Bruce A. C. Cree, Stephen L. Hauser, Roland G. Henry, Ari J. Green.

**Writing – review & editing:** Eduardo Caverzasi, Christian Cordano, Alyssa H. Zhu, Chao Zhao, Antje Bischof, Daniel J. Bennett, Michael Devereux, Nicholas Baker, Nico Papinutto, Jeffrey M. Gelfand, Bruce A. C. Cree, Stephen L. Hauser, Roland G. Henry, Ari J. Green.

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
