## [Decision Letter · Decision Letter 0]

7 Apr 2020

PONE-D-20-00559

Imaging correlates of visual function in Multiple Sclerosis

PLOS ONE

Dear Dr. Green,

Thank you for submitting your manuscript to PLOS ONE. After careful consideration, we feel that it has merit but does not fully meet PLOS ONE’s publication criteria as it currently stands. Therefore, we invite you to submit a revised version of the manuscript that addresses the points raised during the review process.

First, I would like to thank you for your patience during the first round of revisions; it's was very much appreciated.

While there are a number of issues to address from both Reviewers, most of them are relatively minor; both suggested that only a minor revision is required. Regarding the point raised by Reviewer 1 and the reporting of the OCT data with respect to the APOSTEL guidelines: I would prefer that this is not dropped, but rather the full information is reported as per these guidelines.

We would appreciate receiving your revised manuscript by May 22 2020 11:59PM. To enhance the reproducibility of your results, we recommend that if applicable you deposit your laboratory protocols in protocols.io, where a protocol can be assigned its own identifier (DOI) such that it can be cited independently in the future. For instructions see: http://journals.plos.org/plosone/s/submission-guidelines#loc-laboratory-protocols

We look forward to receiving your revised manuscript.

Kind regards,

Niels Bergsland

Academic Editor

PLOS ONE

1. Please provide additional details regarding participant consent. In the Methods section, please ensure that you have specified what type of consent you obtained (for instance, written or verbal) and whether the ethics committee approved this consent procedure. If verbal consent was obtained please state why it was not possible to obtain written consent and how verbal consent was recorded. If your study included minors, state whether you obtained consent from parents or guardians.

Reviewers' comments:

Reviewer's Responses to Questions

**Comments to the Author**

1. Is the manuscript technically sound, and do the data support the conclusions?

Reviewer #1: Yes

Reviewer #2: Yes

2. Has the statistical analysis been performed appropriately and rigorously? 

Reviewer #1: Yes

Reviewer #2: Yes

3. Have the authors made all data underlying the findings in their manuscript fully available?

Reviewer #1: No

Reviewer #2: No

4. Is the manuscript presented in an intelligible fashion and written in standard English?

Reviewer #1: Yes

Reviewer #2: Yes

5. Review Comments to the Author

Reviewer #1: In this original research article entitled “Imaging correlates of visual function in Multiple Sclerosis” by Caverzasi et al, a comprehensive structural analysis of the afferent visual pathway was performed, in order to identify the measures that best reflect visual disability in patients with MS. The inclusion of multiple imaging measures, including retinal OCT metrics (pRNFL and GCL) as well as MRI measures of myelin- and neuroaxonal loss along the visual pathway (in thalamus, optic radiations, V1 cortex) is the most important advantage of this study. Moreover, low-contrast letter acuity was used as a more sensitive measure than high-contrast LA, to reflect visual disability in MS.

Patients with previous (recent, i.e. between 5 years and 6 months prior to baseline) ON and patients without clinical ON episodes were included in the study. These two groups were surprisingly were very similar regarding their visual disability.

The results suggested that several measures along the visual pathway (not only retinal OCT-metrics) are associated with visual dysfunction both in patients with- and without previous ON. While in patients with previous ON measures reflecting myelin loss along the visual pathway (in thalamus, and in OR) were most informative reg. visual dysfunction, in patients without previous ON measures of neuroaxonal loss (in the retina and in thalamus and V1) as well as OR lesions seemed most relevant for visual dysfunction.

I find these results very interesting and the article appropriate to be published in the journal. However, I have some questions/suggestions for the authors:

Abstract:

-Please define NODDI

- “Measures of neurodegeneration described most of the observed variance in visual disability”. I would propose to also mention that these neurodegenerative measures were along the entire visual pathway (from retina to V1), if the restriction in word count allows.

- I would suggest to change the sentence: “Absent clinically identifiable inflammatory episodes, residual disability correlates with neurodegeneration…” to : “In absence of clinically identifiable….”

Methods

- Page 5: The average (SD) EDSS is given; I would suggest to use the median (range) instead, due to the non-normal distribution of this ordinal measure.

- It would be helpful to have the descriptive statistics (demographic characteristics of patients, ON/no ON, time since ON, EDSS etc.) summarized in a table, if possible. Moreover, it should be briefly mentioned if all patients had RRMS or if also patients with progressive MS were included in this study.

- Page 5: “….was an exclusion criteria for this study.”: Please correct to: “ was an exclusion criterion for this study”

- Please briefly explain the rationale of excluding patients with ON more than 5 years ago

- Page 6: “An expert neuroradiologist (EC) assessed the number of occipital, cortical lesions and also segmented the total, white-matter-lesion burden within the optic radiation, using the available FLAIR and T1-weigthed 3D MPRAGE images». Could you please briefly explain how this was done? Were the occipital lobe and the OR localized by merely the “eyes” of the rater or you used a combination of the FLAIR-lesion masks and the V1-Free-surfer segmentation/the atlas-based OR VOI to confirm that lesions were indeed part of the V1/the OR? If the first was the case, I would briefly mention this “manual” character of the analysis as a minor limitation in the discussion.

OCT:

- Please briefly explain why GCL was used instead of GCIPL (since the latter is used more widely, due to data suggesting that the segmentation of the GCIPL is more reproducible than of the GCL alone).

- Was the GCL averaged between both eyes for the models? (also for ON+ patients?)

- It is stated, that: “We followed the APOSTEL guidelines for reporting OCT studies”. However, no details reg. the OCT protocol, excluded scans etc. are provided in the paper. I would either delete this sentence, or provide all the details according to “APOSTEL” as supplementary material.

- Due to the large amount of metrics used in the study, many of which do not appear in the results, if they were not predictive enough in the models, I believe that a Table summarizing all the metrics used (from retina to V1) and their mean values at baseline, would be helpful for the reader (perhaps as supplementary material?). Alternatively, I would at least suggest to specifically name the metrics that later appear in the results (such as “thalamic myelin” etc.) also in the Methods (e.g. In the sentence: “two-shell,neurite-orientation dispersion and density imaging (NODDI) [12] protocol (30 & 64 directions at b = 700 & 2000 s/mm2, 2.2 mm3 cubic voxel) and a multi-echo gradient-echo (MEGE) sequence “for myelin content quantification…” you could add which metrics derived from this method (e.g.: “ to assess the average thalamic myelin” etc.)

Results

In general, since especially in patients without ON many measures might reflect global CNS neurodegeneration rather than damage restricted in the visual pathway, it may be useful to include (i.e. to correct for) normalised brain volume in the models.

- Page 7: Similarly to my comment above reg. the EDSS, I would assume that EDSS and disease duration are probably not normally distributed, thus better the median (range) instead of mean (SD) should be provided.

- “Subjects with no history of ON showed an average low contrast letter acuity (LCLA) score of 23 ± 10 SD. Subjects with a history of ON scored an average of 23 letters ± 9 SD on LCLA testing”.:

It is surprising for the reader, that patients with ON do not have worse LCLA than patients without ON. You did not mention in “Methods” weather LCLA was measured mono- or binocularly, but I suppose (seeing these results) the latter was the case. Please clarify this in “Methods”.

Moreover, it should be mentioned in “Methods” or “Results” if ON was unilateral in all cases. If this was indeed the case, wouldn’t it be preferable to perform the LCLA ON+ model only for the affected eye (if of course data from mono-ocular LCLA testing are available)?

In such a model the LCLA of the affected eye would be the DV, similarly the OCT parameters only from the affected eye would be included in this model, while the other measures (OR, thalamus, V1) would remain as they are.

I would expect that in such a model unilateral GCL thickness would be a better predictor as in the current ON+ model (currently surprising for me that GCL is a better predictor of visual acuity in the ON- vs. the ON+ model).

Page 8: “Measures of neurodegeneration (e.g. volumes of gray matter structures) seemed equally informative in both negative and positive history of optic neuritis, contributing to clinical disability.”. In table 1, thalamic volumes appear only to be important for patients with negative ON history. Moreover, V1 Volume is mentioned once as “V1 GM volume” (in the ON- model) and once “V1 VOIs volumes” (in the ON+ model) is this the same parameter? If yes, I would reformulate the above sentence, e.g.:

“The volume of cortical gray matter structures (particularly V1) seemed equally informative in both negative and positive history of optic neuritis, while thalamic volume was more informative in patients without ON.”

However, it should be noted, that the scale coefficient for “V1 VOIs volumes” in the ON+ model is negative, which –to my understanding- would mean that the thicker the V1 cortex the worse the vision. Please confirm and/or comment.

Were the DTI-measures of the OR (mean MD) less informative than the OR-Myelin in the ON+ model?

Discussion

In general, I believe that in such an analysis, using several measures of the visual pathway, which are not independent from each other (e.g. the association of thalamic- and OR-myelin with LCLA in the ON+ model might well be due to an anterograde transsynaptic degeneration after ON thus these measures may be well correlated with GCL thickness in these patients), the interpretation of the results should be made with more caution. I propose to briefly comment on this in the discussion.

- As also mentioned above I find it surprising that GCL is less informative than e.g. thalamic myelin in patients with positive ON history, since several previous papers showed GCIPL to be the best-correlated measure with LCLA, visual quality of life etc, in patients with MS-ON. I believe that the authors should comment on this (and the possible reasons) in the discussion (see also my comment above reg. monocular vs. binocular LCLA testing).

- It is mentioned that: “neurodegeneration may follow myelin loss by a prolonged period - even years- and therefore functional impairment from an episode in the more distant past may be more likely to be related to neuroaxonal loss).”

If this would explain the results of the current paper, it would also imply that patients with negative ON history in this study may have had an ON episode in the more distant past. Can this be the case? (I understood from Methods and exclusion criteria that patients with ON never had ON). Alternatively, please re-formulate this sentence.

- Having said this, I believe that another explanation for the differences between the ON- and ON+ models might be that damage in the posterior pathway (OR lesions, V1 neurodegeneration) may contribute to visual disability particularly in patients without ON (OR lesion volume remained only in the ON- model, V1 Volume is more informative in the ON- model). Please briefly discuss. Moreover, the sentence: “By contrast, in patients without prior episodes of optic neuritis, only axonal neurodegeneration appears to be responsible for clinical disability.”, is not completely true, since also OR lesion volume, which is not directly a neurodegenerative measure seems relevant in these patients.

- As weakness of this study it is mentioned that : “some of the data may have been influenced by the criteria for inclusion in the clinical trial [9]. For example, patients with mild visual disability or dysfunction were likely excluded.” Please also briefly comment on excluded patients that had ON > 5 years ago and on patients with progressive MS (if indeed excluded).

Further, I believe that the lack of healthy controls is a drawback, since it cannot be concluded which metrics were indeed abnormal compared to HC and also the grade of visual disability of the patients cannot be directly appreciated. I propose to mention this in the discussion and also, if possible, to add whether the included patients could be seen as visually impaired (according to indirect comparison of their LCLA results with LCLA values of healthy controls in the literature).

Reviewer #2: The aim of this work was to describe visual impairment in MS, using a multimodal approach, combining quantitative MRI (qMRI) techniques and OCT.

Clinical, qMRI and OCT data comes from the baseline assessments of the ReBUILD study, a double blind, placebo-controlled phase 2 trial that showed the efficacy of Clemastine in improving VEP latencies and visual function, by stimulating remyielination.

The design and methodology, as well as, results of the study are clear and sufficiently exhaustive.

I have just a few minor comments:

Abstract: the sentence "absent clinical..." is not clear and should be rephrased

Methods:

1) how the authors managed possible/expected parcellation errors made by Freesurfer?

2) I'm a little surprised to see that ON+ patients had the same LCLA score as ON- subjects. In order to better characterize the two subgroups of subjects, the authors might also briefly show VEP latencies and OCT results.

3) if permitted by the publisher, I'd like to see a table reporting, for each group of subjects, qMRI and OCT measured metrics that were included in the predictive models.

4) since LCLA scores were substantially equal in the two subgroups of subjects, did the authors look at a LCLA model including the whole population? I think it would be interesting to look also at the results of such an analysis.

Results/Discussion:

I would suggest underpinning in the discussion that in order to be applied in observational and clinical studies, the (obtained) predictive model needs to be replicated in other/larger populations and, possibly, simplified.

6. PLOS authors have the option to publish the peer review history of their article (what does this mean?). If published, this will include your full peer review and any attached files.

Reviewer #1: No

Reviewer #2: No

---

## [Author Response · Author response to Decision Letter 0]

28 May 2020

Please see response to review document

Reviewer #1

In this original research article entitled “Imaging correlates of visual function in Multiple Sclerosis” by Caverzasi et al, a comprehensive structural analysis of the afferent visual pathway was performed, in order to identify the measures that best reflect visual disability in patients with MS. The inclusion of multiple imaging measures, including retinal OCT metrics (pRNFL and GCL) as well as MRI measures of myelin- and neuroaxonal loss along the visual pathway (in thalamus, optic radiations, V1 cortex) is the most important advantage of this study. Moreover, low-contrast letter acuity was used as a more sensitive measure than high-contrast LA, to reflect visual disability in MS.

Patients with previous (recent, i.e. between 5 years and 6 months prior to baseline) ON and patients without clinical ON episodes were included in the study. These two groups were surprisingly very similar regarding their visual disability. The results suggested that several measures along the visual pathway (not only retinal OCT-metrics) are associated with visual dysfunction both in patients with- and without previous ON. While in patients with previous ON measures reflecting myelin loss along the visual pathway (in thalamus, and in OR) were most informative reg. visual dysfunction, in patients without previous ON measures of neuroaxonal loss (in the retina and in thalamus and V1) as well as OR lesions seemed most relevant for visual dysfunction. I find these results very interesting and the article appropriate to be published in the journal. However, I have some questions/suggestions for the authors:

A: We thank the reviewer for his/her positive comments.

Abstract:

-Please define NODDI

- “Measures of neurodegeneration described most of the observed variance in visual disability”. I would propose to also mention that these neurodegenerative measures were along the entire visual pathway (from retina to V1), if the restriction in word count allows.

- I would suggest to change the sentence: “Absent clinically identifiable inflammatory episodes, residual disability correlates with neurodegeneration…” to : “In absence of clinically identifiable….”

A: We modified the abstract based on the excellent suggested edits – including defining NODDI.

Methods

- Page 5: The average (SD) EDSS is given; I would suggest to use the median (range) instead, due to the non-normal distribution of this ordinal measure.

A: We now report the EDSS as median (range)

- It would be helpful to have the descriptive statistics (demographic characteristics of patients, ON/no ON, time since ON, EDSS etc.) summarized in a table, if possible. Moreover, it should be briefly mentioned if all patients had RRMS or if also patients with progressive MS were included in this study.

A: We thank the reviewer for his/her comment. We now mention that all the subject enrolled in the study had RRMS. We also added a table (now Table 1) summarizing the demographics and clinical information for the two groups (negative and positive history of optic neuritis).

- Page 5: “….was an exclusion criteria for this study.”: Please correct to: “ was an exclusion criterion for this study”.

A: We have edited accordantly. 

- Please briefly explain the rationale of excluding patients with ON more than 5 years ago

A: We thank the reviewer for this comment. This was an exclusion criterion for the REBUILD trial and we appreciate the need for clarification in the current manuscript. We modified accordingly. For the REBUILD trial, the “qualifying eye” could not have had an ON more than 5 years ago because the purpose of the trial was to induce remyelination using clemastine as a remyelinating treatment. We considered lesions older than 5 years to have less remyelinating potential. However both eyes were examined, even the eyes with ON more than 5 years prior the study and we utilized all data for the current analysis. We now mention this in the Methods (line 48 of the revised manuscript with track changes).

- Page 6: “An expert neuroradiologist (EC) assessed the number of occipital, cortical lesions and also segmented the total, white-matter-lesion burden within the optic radiation, using the available FLAIR and T1-weigthed 3D MPRAGE images». Could you please briefly explain how this was done? Were the occipital lobe and the OR localized by merely the “eyes” of the rater or you used a combination of the FLAIR-lesion masks and the V1-Free-surfer segmentation/the atlas-based OR VOI to confirm that lesions were indeed part of the V1/the OR? If the first was the case, I would briefly mention this “manual” character of the analysis as a minor limitation in the discussion.

A: We thank the reviewer to highlight the missing information. We did use the Freesurfer and JHU atlas based mask to confirm the location the cortical and white matter lesions. We addressed this in the Methods (line 111 of the revised manuscript with track changes).

OCT:

- Please briefly explain why GCL was used instead of GCIPL (since the latter is used more widely, due to data suggesting that the segmentation of the GCIPL is more reproducible than of the GCL alone).

A: Thanks for raising this important point. The rationale for the use of GCIPL is based on the reported failure of intra-retinal layer segmentation of macular OCT when distinguishing the boundary between GCL and IPL. This concept is true for some devices, however with more recent software updates the Heidelberg Spetralis is successful in distinguishing these 2 layers. We recently performed a test-retest study on 20 subjects in a separate project (unpublished data available on request) showing no meaningful differences in terms of coefficient of variance between GCL, IPL and INL. As a consequence, we utilized GCL as layer which reflects cell soma of the RGCs. 

- Was the GCL averaged between both eyes for the models? (also for ON+ patients?)

A: This is correct. GCL (as all the other measures) were averaged between both eyes

 It is stated, that: “We followed the APOSTEL guidelines for reporting OCT studies”. However, no details reg. the OCT protocol, excluded scans etc. are provided in the paper. I would either delete this sentence, or provide all the details according to “APOSTEL” as supplementary material.

A: We now prepared an “APOSTEL” table included in the supplemental material.

- Due to the large amount of metrics used in the study, many of which do not appear in the results, if they were not predictive enough in the models, I believe that a Table summarizing all the metrics used (from retina to V1) and their mean values at baseline, would be helpful for the reader (perhaps as supplementary material?). Alternatively, I would at least suggest to specifically name the metrics that later appear in the results (such as “thalamic myelin” etc.) also in the Methods (e.g. In the sentence: “two-shell,neurite-orientation dispersion and density imaging (NODDI) [12] protocol (30 & 64 directions at b = 700 & 2000 s/mm2, 2.2 mm3 cubic voxel) and a multi-echo gradient-echo (MEGE) sequence “for myelin content quantification…” you could add which metrics derived from this method (e.g.: “ to assess the average thalamic myelin” etc.)

A: We thank again the reviewer for her/his comment. In addition to conventional MRI metrics (volumes and lesion burden), we decided to focus the analysis on three specific metrics: MD (DTI derived), ODI (NODDI based) and MWF specific for myelin content (MEGE sequence). We now reported this more clearly in section “MRI analysis” of the Methods (line 89 of the revised manuscript with track changes) and we now report more clearly within the entire manuscript that myelin water fraction (MWF) was used as putative myelin metric. In the Supplementary material we now report each metric (OCT or MRI) mean values.

Results

In general, since especially in patients without ON many measures might reflect global CNS neurodegeneration rather than damage restricted in the visual pathway, it may be useful to include (i.e. to correct for) normalised brain volume in the models.

A: The authors agree on this point. All brain volumes were normalized (using total intracranial volume) (line 83 of the revised manuscript with track changes). 

- Page 7: Similarly to my comment above reg. the EDSS, I would assume that EDSS and disease duration are probably not normally distributed, thus better the median (range) instead of mean (SD) should be provided.

A: We now report EDSS, disease duration and time from ON as median (range).

- “Subjects with no history of ON showed an average low contrast letter acuity (LCLA) score of 23 ± 10 SD. Subjects with a history of ON scored an average of 23 letters ± 9 SD on LCLA testing”.:

It is surprising for the reader, that patients with ON do not have worse LCLA than patients without ON. You did not mention in “Methods” weather LCLA was measured mono- or binocularly, but I suppose (seeing these results) the latter was the case. Please clarify this in “Methods”.

A: The reason for the similar LCLA results within the two groups is due to the inclusion criteria of the Rebuild trial. To be included the patients needed to have a visual deficit (i.e. VEP p100 > 118 ms in at least one eye). The reported LCLA data were measured monocularly and averaged for the model analysis. We now clarified this in Methods (line 52 of the revised manuscript with track changes).

Moreover, it should be mentioned in “Methods” or “Results” if ON was unilateral in all cases. If this was indeed the case, wouldn’t it be preferable to perform the LCLA ON+ model only for the affected eye (if of course data from mono-ocular LCLA testing are available)?

In such a model the LCLA of the affected eye would be the DV, similarly the OCT parameters only from the affected eye would be included in this model, while the other measures (OR, thalamus, V1) would remain as they are.

I would expect that in such a model unilateral GCL thickness would be a better predictor as in the current ON+ model (currently surprising for me that GCL is a better predictor of visual acuity in the ON- vs. the ON+ model).

A: This is an interesting point. We agree with the reviewer about the expected results. We didn’t perform this analysis because our cohort contained patients with bilateral ON.

Page 8: “Measures of neurodegeneration (e.g. volumes of gray matter structures) seemed equally informative in both negative and positive history of optic neuritis, contributing to clinical disability.”. In table 1, thalamic volumes appear only to be important for patients with negative ON history. Moreover, V1 Volume is mentioned once as “V1 GM volume” (in the ON- model) and once “V1 VOIs volumes” (in the ON+ model) is this the same parameter? If yes, I would reformulate the above sentence, e.g.:

“The volume of cortical gray matter structures (particularly V1) seemed equally informative in both negative and positive history of optic neuritis, while thalamic volume was more informative in patients without ON.”

A: We modified the sentence following the reviewer’s comment: “Measures of cortical neurodegeneration seemed equally informative in both negative and positive history of optic neuritis, contributing to clinical disability. Thalamic volume was more informative in patients without ON” (line 158 of the revised manuscript with track changes). We now consistently refer to V1 volume both in the text as well as in tables.

However, it should be noted, that the scale coefficient for “V1 VOIs volumes” in the ON+ model is negative, which –to my understanding- would mean that the thicker the V1 cortex the worse the vision. Please confirm and/or comment.

A: We thank the reviewer for his comment. The V1 volume scale coefficient showed opposite direction (positive in ON- history and negative in ON+ history). This “predictor” of the LCLA models survived also the cross-validation (survived also in LASSO, cross validation is a method to test model RMSE stability). This apparent inconsistency can be related to the nature of the PLS analysis and how the model is constructed. We have to consider the predictor as part of the model rather than one single variable predicting LCLA. It could represent a correction within the model for the observed contribution of the other variables to LCLA, specifically the model in ON+ history was overestimating the V1 contribution to the model.

Were the DTI-measures of the OR (mean MD) less informative than the OR-Myelin in the ON+ model?

A: Absolutely. DTI derived MD was not selected in either models and in particular in the ON+ one, whereas thalamic and OR myelin derived measure (MWF) were the most informative.

Discussion

In general, I believe that in such an analysis, using several measures of the visual pathway, which are not independent from each other (e.g. the association of thalamic- and OR-myelin with LCLA in the ON+ model might well be due to an anterograde transsynaptic degeneration after ON thus these measures may be well correlated with GCL thickness in these patients), the interpretation of the results should be made with more caution. I propose to briefly comment on this in the discussion.

- As also mentioned above I find it surprising that GCL is less informative than e.g. thalamic myelin in patients with positive ON history, since several previous papers showed GCIPL to be the best-correlated measure with LCLA, visual quality of life etc, in patients with MS-ON. I believe that the authors should comment on this (and the possible reasons) in the discussion (see also my comment above reg. monocular vs. binocular LCLA testing).

A: We thank the reviewer for his comment. PLS analysis should limit collinearity across the studied features. Moreover a cross-validation analysis (LASSO) was also performed to test the validity of the identify “predictors” of LCLA in both models and 9 out of 10 (as reported in the manuscripts) predictors were confirmed. Regarding the contribution of GCL we now added in the supplementary material the model that includes the entire cohort. Across models GCL is the most informative. It is true that there appears to be less “informative” power in the ON+ history compared to MWF measurements. As previously stated, we need to look at these predictors within the model and as part of the model. It may be that part of the informative contribution of GCL was “addressed” already by other metrics (e.g. cortical GM and V1 volumes). 

- It is mentioned that: “neurodegeneration may follow myelin loss by a prolonged period - even years- and therefore functional impairment from an episode in the more distant past may be more likely to be related to neuroaxonal loss).”

If this would explain the results of the current paper, it would also imply that patients with negative ON history in this study may have had an ON episode in the more distant past. Can this be the case? (I understood from Methods and exclusion criteria that patients with ON never had ON). Alternatively, please re-formulate this sentence.

A: The reviewer understood correctly. Patients without ON never had clinically evident ONs. But the inclusion criteria for the REBUILD clinical trial were designed to ensure that patients had demyelinating injury in the visual pathway (i.e. VEP P100 latency in at least one eye of 118 ms). For this reason, these patients likely presented paucisymptomatic inflammatory episodes within the visual pathway. We changed the sentence in the Discussion (line 195 of the revised manuscript with track changes)..

- Having said this, I believe that another explanation for the differences between the ON- and ON+ models might be that damage in the posterior pathway (OR lesions, V1 neurodegeneration) may contribute to visual disability particularly in patients without ON (OR lesion volume remained only in the ON- model, V1 Volume is more informative in the ON- model). Please briefly discuss. Moreover, the sentence: “By contrast, in patients without prior episodes of optic neuritis, only axonal neurodegeneration appears to be responsible for clinical disability.”, is not completely true, since also OR lesion volume, which is not directly a neurodegenerative measure seems relevant in these patients.

A: Thanks for your comment. We modified the sentence in the Discussion (line 178 of the revised manuscript with track changes).

- As weakness of this study it is mentioned that : “some of the data may have been influenced by the criteria for inclusion in the clinical trial [9]. For example, patients with mild visual disability or dysfunction were likely excluded.” Please also briefly comment on excluded patients that had ON > 5 years ago and on patients with progressive MS (if indeed excluded).

A: The authors believe that having clarified the sentence about the exclusion criteria this cannot be considered a limitation of the work. We added the disease course of the patients included in the trial in methods.

Further, I believe that the lack of healthy controls is a drawback, since it cannot be concluded which metrics were indeed abnormal compared to HC and also the grade of visual disability of the patients cannot be directly appreciated. I propose to mention this in the discussion and also, if possible, to add whether the included patients could be seen as visually impaired (according to indirect comparison of their LCLA results with LCLA values of healthy controls in the literature).

A: The authors agree on this point. Unfortunately, given the study design, no healthy controls were analyzed for this study. In this recent publication, “Temporal visual resolution and disease severity in MS” by Noah Ayadi, et al. (Neurol Neuroimmunol Neuroinflamm 2018), the average Sloan 2.5% score for 62 healthy control eyes was 45.48±11.22. We added this paper in the bibliography.

 

Reviewer #2

The aim of this work was to describe visual impairment in MS, using a multimodal approach, combining quantitative MRI (qMRI) techniques and OCT.

Clinical, qMRI and OCT data comes from the baseline assessments of the ReBUILD study, a double blind, placebo-controlled phase 2 trial that showed the efficacy of Clemastine in improving VEP latencies and visual function, by stimulating remyielination.

The design and methodology, as well as, results of the study are clear and sufficiently exhaustive.

A: We thank the reviewer for his/her positive comments.

I have just a few minor comments:

Abstract: the sentence "absent clinical..." is not clear and should be rephrased

A: We addressed this in the abstract.

Methods:

1) how the authors managed possible/expected parcellation errors made by Freesurfer?

A: Freesurfer pipeline segmentation and parcellation was visually checked by an expert neuroradiologist. 

2) I'm a little surprised to see that ON+ patients had the same LCLA score as ON- subjects. In order to better characterize the two subgroups of subjects, the authors might also briefly show VEP latencies and OCT results.

A: We now report VEP and OCT results of the two subgroups in the Supplementary material.

3) if permitted by the publisher, I'd like to see a table reporting, for each group of subjects, qMRI and OCT measured metrics that were included in the predictive models.

A: We now added this data in Supplementary material.

4) since LCLA scores were substantially equal in the two subgroups of subjects, did the authors look at a LCLA model including the whole population? I think it would be interesting to look also at the results of such an analysis.

A: We now looked at the entire population and reported in the Supplementary material.

Results/Discussion:

I would suggest underpinning in the discussion that in order to be applied in observational and clinical studies, the (obtained) predictive model needs to be replicated in other/larger populations and, possibly, simplified.

A: We now included this comment in the discussion (line 209 of the revised manuscript with track changes).

---

## [Decision Letter · Decision Letter 1]

19 Jun 2020

Imaging correlates of visual function in Multiple Sclerosis

PONE-D-20-00559R1

Dear Dr. Green,

We’re pleased to inform you that your manuscript has been judged scientifically suitable for publication and will be formally accepted for publication once it meets all outstanding technical requirements.

Kind regards,

Niels Bergsland

Academic Editor

PLOS ONE

Additional Editor Comments (optional):

Reviewers' comments:

Reviewer's Responses to Questions

**Comments to the Author**

1. If the authors have adequately addressed your comments raised in a previous round of review and you feel that this manuscript is now acceptable for publication, you may indicate that here to bypass the “Comments to the Author” section, enter your conflict of interest statement in the “Confidential to Editor” section, and submit your "Accept" recommendation.

Reviewer #1: All comments have been addressed

2. Is the manuscript technically sound, and do the data support the conclusions?

Reviewer #1: Yes

3. Has the statistical analysis been performed appropriately and rigorously? 

Reviewer #1: Yes

4. Have the authors made all data underlying the findings in their manuscript fully available?

Reviewer #1: No

5. Is the manuscript presented in an intelligible fashion and written in standard English?

Reviewer #1: Yes

6. Review Comments to the Author

Reviewer #1: I thank the authors for addressing all comments very well. I have no further comments. I believe that the statistical analysis was performed appropriately, the paper is well written and the results interesting. For my understanding the authors do not make all data fully available without restriction. I have no further comments.

7. PLOS authors have the option to publish the peer review history of their article (what does this mean?). If published, this will include your full peer review and any attached files.

Reviewer #1: Yes: Athina Papadopoulou

---

## [Editor Report · Acceptance letter]

22 Jul 2020

PONE-D-20-00559R1 

Imaging correlates of visual function in Multiple Sclerosis 

Dear Dr. Green:

I'm pleased to inform you that your manuscript has been deemed suitable for publication in PLOS ONE. Congratulations! Your manuscript is now with our production department. 

Kind regards, 

on behalf of

Dr. Niels Bergsland 

Academic Editor

PLOS ONE